# Identification of and Mechanistic Insights into SARS-CoV-2 Main Protease Non-Covalent Inhibitors: An In-Silico Study

**DOI:** 10.3390/ijms24044237

**Published:** 2023-02-20

**Authors:** Jian-Xin Shen, Wen-Wen Du, Yuan-Ling Xia, Zhi-Bi Zhang, Ze-Fen Yu, Yun-Xin Fu, Shu-Qun Liu

**Affiliations:** 1State Key Laboratory for Conservation and Utilization of Bio-Resources in Yunnan, School of Life Sciences, Yunnan University, Kunming 650091, China; 2Yunnan Key Laboratory of Stem Cell and Regenerative Medicine, Biomedical Engineering Research Center, Kunming Medical University, Kunming 650500, China; 3Human Genetics Center and Department of Biostatistics and Data Science, School of Public Health, The University of Texas Health Science Center, Houston, TX 77030, USA

**Keywords:** SARS-CoV-2 Mpro, non-covalent inhibitors, binding affinity, protein-ligand interactions, virtual screening

## Abstract

The indispensable role of the SARS-CoV-2 main protease (Mpro) in the viral replication cycle and its dissimilarity to human proteases make Mpro a promising drug target. In order to identify the non-covalent Mpro inhibitors, we performed a comprehensive study using a combined computational strategy. We first screened the ZINC purchasable compound database using the pharmacophore model generated from the reference crystal structure of Mpro complexed with the inhibitor ML188. The hit compounds were then filtered by molecular docking and predicted parameters of drug-likeness and pharmacokinetics. The final molecular dynamics (MD) simulations identified three effective candidate inhibitors (ECIs) capable of maintaining binding within the substrate-binding cavity of Mpro. We further performed comparative analyses of the reference and effective complexes in terms of dynamics, thermodynamics, binding free energy (BFE), and interaction energies and modes. The results reveal that, when compared to the inter-molecular electrostatic forces/interactions, the inter-molecular van der Waals (vdW) forces/interactions are far more important in maintaining the association and determining the high affinity. Given the un-favorable effects of the inter-molecular electrostatic interactions—association destabilization by the competitive hydrogen bond (HB) interactions and the reduced binding affinity arising from the un-compensable increase in the electrostatic desolvation penalty—we suggest that enhancing the inter-molecular vdW interactions while avoiding introducing the deeply buried HBs may be a promising strategy in future inhibitor optimization.

## 1. Introduction

Severe acute respiratory syndrome coronavirus 2 (SARS-CoV-2), a new coronavirus causing coronavirus diseases 2019 (COVID-19), was first reported in Wuhan, China, in late December, 2019 [1] and caused a global pandemic only four months after its first appearance [2]. As of December 2022, there had been more than 661 million confirmed cases of COVID-19, including 6.7 million deaths (http://www.who.int/data (accessed on 16 January 2023)). Even worse, SARS-CoV-2 keeps evolving into new variants with increased transmissibility and reduced neutralization by antibodies generated against previous infections or vaccinations [3,4,5], thus posing a persistent threat to the public’s health and the global economy. Although there are several approved drugs for the treatment of COVID-19 (https://www.fda.gov/consumers/consumer-updates/know-your-treatment-options-covid-19 (accessed on 9 January 2023)), continuous efforts are still required to develop/find effective drugs against SARS-CoV-2, its variants, previous epidemical coronaviruses (CoVs), or even broader CoVs.

Similar to other CoVs, the SARS-CoV-2 is a single-stranded positive-sense RNA virus, with its genome coding for four structural proteins and 16 non-structural proteins (Nsps) [6]. Upon entry into the host cell, the two overlapping open reading frames (ORFs), ORF1a and ORF1b, are translated into two polyproteins, pp1a and pp1ab [7,8]. Subsequently, the two conserved viral proteases, main protease (Mpro), also known as 3-chymotrypsin-like cysteine protease (3CLpro), and papain-like protease (Plpro), proteolytically cleave the two polyproteins to generate 16 Nsps [9,10,11]. These Nsps, together with some host factors, assemble into a replication transcription complex that is required for viral genome replication and the production of new viruses [8,12,13,14,15]. Therefore, inhibiting the activities of Mpro and PLpro could be an effective strategy for blocking the CoV life cycle. However, the PLpro seems not to be an ideal target because it also recognizes the C-terminal sequence of ubiquitin, and hence its inhibitors would be expected to also inhibit the host-cell deubiquitinases [16]. In contrast, the CoV Mpro exclusively cleaves the peptide bond between the glutamine at position P1 and a small amino acid (serine, alanine, or glycine) at position P1′, thus indicative of distinctly different substrate specificity from the human host-cell proteases [17,18,19]. Furthermore, both the sequence and structure of Mpro are highly conserved among different CoVs [17,18], and the SARS-CoV-2 Mpro was estimated to have a lower non-synonymous mutation rate than the spike protein and RNA-dependent RNA polymerase [20]. Collectively, Mpro is thought of as one of the most ideal viral targets for the development/discovery of broad-spectrum antivirals against COVID-19 and other coronavirus diseases [21,22].

The SARS CoV Mpro (EC 3.4.22.69), formally known as C30 endopeptidase, is a cysteine protease that belongs to the PA protease clan. The active form of Mpro is a homodimer of identical protomers/monomers (Nsp5), which derives from the auto-cleavage between Nsp4 and Nsp6 [23,24,25]. Mpro has a cysteine-histidine catalytic dyad (Cys145-His41 in the case of the SARS-CoV-2 Mpro) at its active center [26,27], in which the cysteine side-chain sulfydryl acts as a nucleophile and the histidine imidazole ring as a general base and proton acceptor [18,28,29].

As shown in Figure 1A, the functional form of the SARS-CoV-2 Mpro [30] is composed of two non-covalently associated protomers. Each protomer (306 amino acids) contains an N-terminal finger (residues 1–7) and three domains (I, II, and III), of which the domain I and II fold as an anti-parallel β-barrel similar to the chymotrypsin-like folding scaffold (Figure 1B). The substrate-binding cavity, located between domains I and II, contains a series of pockets/subsites: S4, S3, S2, S1, and S1′ (Figure 1C,D). Each of them accommodates/binds to a single amino acid residue (P4, P3, P2, P1, and P1′ from the N- to C-terminus) of a consecutive polypeptide substrate or inhibitor [31,32,33]. The catalytic dyad Cys145-His41, which hydrolytically cleaves the peptide bond between P1 and P1′, participates in the formation of the S1′ pocket (Figure 1B,D).

Considerable efforts have led to the identification of certain inhibitors against the SARS-CoV-2 Mpro with desirable inhibitory effects/potencies. The representative inhibitors include Ebselen [33], Boceprevir [35,36,37], GC-376 [35,36], and calpain inhibitors II and XII [35,38]. Encouragingly, an antiviral compound PF-073213321 (nirmatrelvir) from Pfizer, which was optimized from a previously identified inhibitor (PF-00835231) against the SARS-CoV Mpro [39], demonstrates a potent inhibitory effect on the SARS-CoV-2 Mpro through covalently modifying the nucleophilic residue Cys145 with its nitrile warhead [40]. Since the oral antiviral Paxlovid^TM^ (PF-07321332 co-packaged with ritonavir, an HIV-1 protease inhibitor and CYP3A inhibitor that slows down the breakdown of PF-07321332) was found to significantly reduce the risk of hospitalization or death from COVID-19 in clinical trials [41,42,43], it was granted emergency use authorization (EUA) for the treatment of COVID-19 by the Food and Drug Administration (FDA) on 22 December 2021 [44].

Usually, the process of finding a new drug against a chosen target involves experimental high-throughput screening. However, virtual screening (VS) and computer-aided methods play an increasingly vital role in modern drug discovery due to enormous time and cost savings. There have been many studies aiming to screen/identify potential drug compounds targeting the SARS-CoV-2 Mpro using computational methods. For example, using molecular docking to screen the MolPort database followed by MD simulations and binding free energy (BFE) calculations, Ibrahim et al. identified nine natural product compounds with the highest binding affinity to Mpro [45]. Gentile et al. screened the marine natural product database using combined computational techniques of ligand-based pharmacophore modeling, VS, molecular docking, and MD simulation and identified 17 potential Mpro inhibitors [46]. Sayed et al. screened six compounds with high potential as anti-Mpro drug candidates from the microbial natural product database through a combined method of ligand-based and structure-based VS, drug-likeness evaluation, and MD simulations [47]. Through structure-based VS of a filtered database containing the drug-like food compounds, Masand et al. found that spermidine and its two derivatives have a high affinity to Mpro [48]. Using a multi-stage VS protocol, Mandour et al. successfully identified five drugs with promising binding modes to the substrate-binding cavity of the SARS-CoV-2 Mpro from an FDA-approved drug dataset [49]. Through structure-based VS of 124 FDA-approved anti-microbial drugs followed by MD simulations, Mahanta et al. proposed that Viomycin may act as a potential Mpro inhibitor due to its high binding strength and robust binding to the SARS-CoV-2 Mpro [50].

It should be noted that most of the experimentally identified inhibitors act through covalent binding to the catalytic residue Cys145. However, when using the computational methods to screen the non-covalent inhibitors, most studies used the non-covalent docking score of a covalent inhibitor as a reference for affinity comparison and ranking, which may lead to improper or even wrong screening results. Considering the off-target effects of the covalent inhibitors [51,52] and the error-prone evaluation of the affinity of the non-covalent inhibitors, it is necessary to further strengthen the screening and identification of the non-covalent inhibitors of the SARS-CoV-2 Mpro.

In the present study, therefore, we employed the experimentally determined structure of SARS-CoV-2 Mpro in complex with a non-covalent inhibitor, ML188 [34], as the reference to screen and identify the potential non-covalent inhibitors of Mpro. Through pharmacophore-based VS of the ZINC purchasable database [53], followed by pharmacokinetic property predictions, molecular docking, and MD simulations, we obtained three effective candidate inhibitors (ECIs) for further wet-lab validation in a future study. We also elucidated the mechanical and energetic mechanisms underlying the association stability and binding affinity between the identified inhibitors and Mpro, which are expected to provide guidance for modification, optimization, or even the design of the non-covalent inhibitor targeting the SARS-CoV-2 Mpro.

## 2. Results

### 2.1. Pharmacophore and Virtual Screening

Figure 2 shows the pharmacophore model generated by the Pharmit server for the target protein Mpro. This model includes a total of seven pharmacophoric features: three hydrogen bond acceptors (HBAs; orange spheres) located within/near the S1 pocket, one hydrophobic center (HPC; green sphere) near the S1′ pocket, two HPCs within/near the S2 pocket, and one aromatic ring (AR; purple sphere) that overlaps with the HPC near the S2 pocket. The initial VS of the ZINC database using the generated pharmacophore model followed by further filtering with Lipinski’s rule of five (Ro5) yielded a total of 593 hit compounds (Appendix A). The root mean square deviations (RMSDs) between all hits and the pharmacophore model range from 0.36 to 0.78 Å, indicating a high degree of consistency of these hits with the pharmacophore model.

### 2.2. Molecular Docking

The top-ranked pose obtained by re-docking ML188 into the substrate-binding cavity of Mpro has a RMSD value of 0.93 Å relative to the experimental pose. Furthermore, when compared to the co-crystallized ML188, the docked ML188 exhibits an almost consistent number and nature of interactions with Mpro (Appendix A), indicating that AutoDock Vina has the ability to reproduce the experimental binding mode.

The docking score values for the top-ranked poses of the 593 hits against Mpro range from −9.4 to −5.6 kcal/mol (Appendix A). Generally speaking, the lower the score value, the higher the predicted binding affinity of a compound to Mpro. Using the score value of the top-ranked pose of ML188 (−8.1 kcal/mol) as the reference, we screened 78 potential candidate compounds (PCCs) with the score values ≤ −8.1 kcal/mol (Appendix A).

### 2.3. Pharmacokinetic Properties

The 78 PCCs were further filtered by the predicted ADMET parameters. Since the toxicity prediction plays an essential role in reducing the cost and labor of a drug’s preclinical and clinical trials [54], we used the predicted toxicity properties, i.e., AMES mutagenicity, hepatotoxicity, and oral rat acute toxicity (median lethal dose; LD50), as the basic filters. The results reveal that the 78 PCCs had LD50 values ranging from 1.876 to 3.161 mol/kg (Appendix A), exhibiting similar or even lower acute toxicity than aspirin [55,56]. Furthermore, 12 out of the 78 PCCs were predicted to be AMES-positive and hence mutagenic/carcinogenic, and 70 to be hepatotoxic (Appendix A). After removing those with either AMES toxicity or hepatotoxicity, we obtained four ECIs, i.e., ZINC000012406433, ZINC000000930519, ZINC000023909009, and ZINC000000634921, which are referred to as ECI1, ECI2, ECI3, and ECI4, respectively, in the rest of the text. Careful examination of previous studies where the ZINC database (or its sub-dataset) has been screened against the SARS-CoV-2 Mpro [57,58,59,60,61,62,63,64] indicates that the four ECIs have no overlap with previously identified compounds. This is likely due to the different screening strategies/pipelines adopted in our study and previous studies.

Table 1 lists the predicted parameters of ADMET and Ro5 for the four ECIs and ML188. All four ECIs conform to Ro5, as it was employed as a filter in the initial screening phase. However, the reference compound ML188 shows one violation of Ro5, i.e., the octanol-water partition coefficient (logP = 5.27) exceeds the allowed value of five, indicative of its high lipophilicity and bioaccumulation potential. This is consistent with poorer water solubility of ML188 (−4.76) compared to the four ECIs. All ECIs have appreciable human intestinal absorption percentages (far more than 30%), meaning the feasibility of the oral administration route.

In terms of the distribution, the predicted volume of distribution at steady state (VDss) values show that ML188 and ECI4 are more inclined to be distributed in the tissue (log(L/kg) > 0.45), ECI2 is more inclined to be distributed in the blood plasma (log(L/kg) < −0.15), and ECI1 and ECI3 would be distributed relatively evenly in the plasma and tissue (−0.15 < log(L/kg) ≤ 0.45). As for the blood-brain barrier (BBB) permeability, ECI4 and ECI2 were predicted to readily cross the BBB (logBB > 0.3) and be poorly distributed in the brain (logBB < −1.0), respectively, while the rest were predicted to have moderate BBB permeability (−1.0 < logBB < 0.3).

In terms of metabolism, the prediction results show that all ECIs are likely to be substrates or inhibitors of cytochrome P450 3A4 (CYP3A4) rather than of cytochrome P450 2D6 (CYP2D6), implying that they may be metabolized by CYP3A4 or inhibit CYP3A4 activity.

ECI2 and ECI4 were predicted to be potential substrates of renal organic cation transporter 2 (Renal OCT2), implying that Renal OCT2 is likely to be involved in disposition and renal clearance of them. However, ECI1, ECI3, and ML188 were predicted to have a higher total clearance than ECI2 and ECI4. Of note is that the reference compound ML188 was predicted to have hepatotoxicity and AMES toxicity but a lower rat oral acute toxicity (i.e., a higher LD50 value) than the four ECIs.

In summary, all four ECIs were predicted to have satisfactory properties of drug-likeness and pharmacokinetics. In contrast, it seems that the reference compound ML188 is an incompetent drug candidate due to its unfavorable water solubility, carcinogenicity, and hepatotoxicity.

### 2.4. Complex Structural Stability Assessment

The structures of Mpro in complex with ML188 (crystal complex structure) and the four ECIs (docking structures) were subjected to MD simulations to assess their stability. Figure 3 shows the RMSD values of Mpro C_α_ and ligand heavy atoms of the five complexes relative to their respective starting structures. With the exception of Mpro-ECI1, all the other complexes reach equilibrium fluctuations at low RMSD levels after the initial lifting phase. For Mpro-ECI1, its RMSD curve continues to lift and experiences a jump from 0.33 nm at 27 ns to 0.58 nm at 28 ns, after which it reaches equilibrium but fluctuates drastically, especially in the periods of 43–52 and 76–78 ns. Therefore, among the five complexes, Mpro-ECI1 deviated the most from its starting structure while exhibiting the largest structural variability/instability during the simulation. Of interest is the fact that the binding of all five ligands exerts very limited effects on the overall structural dynamics of Mpro because the differences in the fluctuation amplitude of the Mpro C_α_ RMSD curves are minor between the free Mpro and the five ligand-bound Mpro (Appendix A).

Figure 4 shows the structural superposition of snapshots extracted from five different time points in the MD trajectory for each complex. It is clear from Figure 4 that in all complexes, the Mpro structures overlap well with one another, while the locations and conformations of the ligands vary to different extents. The smallest and largest variations in the location and conformation were observed for ML188 and ECI1, respectively. ECI2-4 experience moderate variations in conformation while remaining occupancy at the substrate-binding site/cavity.

Of particular note is that ECI1 “sneaks” away from its initial binding site (0 ns) and reaches a new site located between domains I and II of Mpro at 25 ns (Figure 4B). At 40 and 48 ns, it is located inside and on the outer rim of a shallow groove on the surface of domain II, respectively. However, ECI1 does not escape out of the surface groove because it was observed to be within the groove at the end of the simulation (100 ns), although the bound pose/orientation is inverted compared to that at 40 ns. Therefore, the large changes in RMSD values of Mpro-ECI1 (Figure 3) stem from the large location displacement of ECI1 relative to Mpro.

Since ECI1 escaped out of the substrate-binding site/cavity of Mpro during the simulation, it is unlikely to act efficiently in blocking the substrate binding and hence was excluded from the effective candidate inhibitors. For the other four complexes, their 10–100 ns equilibrated portions of MD trajectories were used for the subsequent analyses.

### 2.5. Interface Residues and Interaction Analysis

Figure 5A shows the identified 29 Mpro residues participating in the formation of the binding interfaces of the four inhibitors. All the interface residues also participate in the formation of the substrate-binding cavity (i.e., the cavity containing the S4, S3, S2, S1, and S1′ pockets/subsites; see Figure 1C,D) with the exception of Thr45 and Val186, which make unique close contacts with ECI2 and ECI4, respectively. With a few exceptions, most of the interface residues make contact with two or more inhibitors. Of note is the fact that the catalytic dyad residues (His41 and Cys145) make contacts with all the four inhibitors. Therefore, it can be considered that the four inhibitors occupy the substrate cavity mainly via interacting with the shared interface residues.

The calculated all-atom root mean square fluctuation (RMSF) values indicate that ML188 (0.90 Å), ECI2 (1.45 Å) and ECI4 (1.50 Å), and ECI3 (2.36 Å) have the lowest, moderate, and highest conformational flexibility/mobility, respectively, in agreement with the differences in the snapshot overlapping degree among them (Figure 4). There are two apparent trends regarding the flexibility of the interface residues: (i) the higher the mobility of an inhibitor, the higher the fraction of its contacting residues with the high all-atom RMSF values and, (ii) the shared interface residues also “share” the flexibility/mobility (i.e., simultaneously high or low RMSF values) in different complexes. Furthermore, the RMSF values averaged over the interface residues of the ML188-, ECI2-, ECI3-, and ECI4-bound Mpro are 0.78, 0.97, 0.99, and 0.89 Å, respectively, suggesting a rough trend that the increased inhibitor mobility is accompanied by the increased mobility of the binding cavity. This may also imply a coordinated change in conformations of both the inhibitor and its binding cavity so as to maintain the protein-ligand association via mutual adaptation.

Figure 5B shows the distributions of the numbers of HBs formed between Mpro and the four inhibitors. In Mpro-ML188 and Mpro-ECI3, the numbers of the inter-molecular HBs range from 0 to 6, and in Mpro-ECI2 and Mpro-ECI4, they range from 0 to 5 and from 0 to 4, respectively. Although the HB number distributions of Mpro-ML188 and Mpro-ECI3 likewise peak at 3 HBs, the former and latter distributions are obviously left-skewed and relatively dispersed, respectively, indicating more stable inter-molecular HBs formed within Mpro-ML188 and fewer, relatively unstable competitive HBs within Mpro-ECI3. The Mpro-ECI2 and Mpro-ECI4 complexes have fewer inter-molecular HBs than Mpro-ML188 and Mpro-ECI3 due to their right-skewed distributions and considerably high percentages of frames carrying no HB. This may also mean that the inter-molecular HB interactions are not of crucial importance in maintaining associations of ECI2 and ECI4 with Mpro.

Figure 5C,D show the boxplot distributions of the inter-molecular vdW and electrostatic interaction energies of the four complexes, respectively. For all the four complexes, their vdW energy medians (Figure 5C) are one order of magnitude lower than the electrostatic energy medians (Figure 5D), indicating that the inter-molecular vdW force is much stronger than the inter-molecular electrostatic force and, hence, contributes substantially to maintaining the protein-inhibitor associations during simulations. Mpro-ML188 and Mpro-ECI2 have similar vdW medians of −189.5 and −188.9 kJ/mol, respectively, which are much lower than that of Mpro-ECI3 (−155.7 kJ/mol) and slightly lower than that of Mpro-ECI4 (−183.1 kJ/mol). Nevertheless, the dispersion of vdW energy distribution follows the order: Mpro-ML188 < Mpro-ECI4 < Mpro-ECI3 < Mpro-ECI2. Therefore, although both Mpro-ML188 and Mpro-ECI2 have the strongest inter-molecular mean vdW forces among the four complexes, the vdW interactions are most stable and unstable in the former and latter complexes, respectively. Another interesting observation is that the variability of the inter-molecular vdW force is approximately proportional to the mobility of the inhibitor-binding cavity. For example, Mpro-ECI2 and Mpro-ECI3 have both higher vdW energy dispersions and inhibitor-binding cavity RMSF values than Mpro-ML188 and Mpro-ECI4.

As for the inter-molecular electrostatic interaction energy distribution (Figure 5D), Mpro-ML188 and Mpro-ECI3 have lower medians but higher dispersions than Mpro-ECI2 and Mpro-ECI4, indicating stronger inter-molecular average electrostatic forces and a larger variability of the electrostatic forces within the two former complexes. This is consistent with the differences in distributions of the inter-molecular HB number as shown in Figure 5B: the higher the percentages of frames with two or more HBs, the stronger the average electrostatic force between the inhibitor and Mpro; the more dispersed the HB number distribution, the larger the variability of the electrostatic force strength. To this end, the changes in the strength and variability of the inter-molecular electrostatic forces likely originate from the changes in the strength and number distribution of HBs formed between the inhibitor and Mpro, although the electrostatic interactions make only a limited contribution to the protein-inhibitor association for all the four complexes.

### 2.6. Free Energy Landscape (FEL)

Figure 6 shows the constructed two-dimensional (2D) FELs of the four complexes as a function of interface RMSD (iRMSD) and the buried surface area between the inhibitor and Mpro. The sizes of the covered regions of the four FELs are in the following order: Mpro-ML188 < Mpro-ECI4 < Mpro-ECI2 < Mpro-ECI3, which reflects the ordering of the magnitude of the interface entropy. The FELs of Mpro-ML188, Mpro-ECI2, and Mpro-ECI4 present funnel-like shapes because of the gradual lowering of free energy without interruption by local minima until reaching the global minimum. In contrast, Mpro-ECI3′s FEL (Figure 6C) contains two large energy basins, between which there are high barriers separating the two minima within the individual basins. Therefore, Mpro-ECI3 sampled two main interface states during the MD simulation. Furthermore, the global minima in FELs of Mpro-ML188 and Mpro-ECI4 have the same energy level, −16 kJ/mol, lower than that of Mpro-ECI2 and Mpro-ECI3, −15 kJ/mol. This indicates the higher interface thermostability or more stable protein-inhibitor association of the former two complexes, in agreement with the earlier comparative analyses in terms of the snapshot superposition, interface residue RMSF, and variability of the intermolecular vdW interactions.

Despite the differences in the interface thermodynamics among the four complexes, the constructed FELs still identify the most thermostable or dominant interface states of the four complexes. As a result, for each complex, 500 snapshots were randomly extracted from the global free energy minimum of FEL and taken as the representative structural ensemble for the subsequent BFE calculation.

### 2.7. Binding Free Energy (BFE)

Table 2 lists the average values and standard deviations (SDs) of BFE and its energy components calculated over the representative structural ensemble of each complex. For the four complexes, the total BFE (Δ*G*_binding_) average values are in the order of Mpro-ECI2 (−136.3 kJ/mol) < Mpro-ML188 (−128.9 kJ/mol) < Mpro-ECI4 (−122.7 kJ/mol) < Mpro-ECI3 (−83.0 kJ/mol). Despite the relatively large SDs of these average values, the unpaired pairwise *t*-tests between all complex groups show that the BFE average values differ significantly for all pairs (the *p*-value ranges from 8.7 × 10^−312^ to 1.0 × 10^−12^). Therefore, it can be concluded that ECI2 and ECI3 have the strongest and weakest binding affinity to Mpro, respectively, whereas ML188 has a significantly stronger Mpro-binding affinity than ECI4.

For all the four complexes, the changes in the vdW interaction energy (Δ*E*_vdW_), electrostatic interaction energy (Δ*E*_elec_), and non-polar solvation free energy (Δ*G*_non-polar_) make positive contributions to lowering the BFE (or increasing binding affinity), whereas the polar desolvation free energy (Δ*G*_polar_) makes a negative contribution to lowering the BFE. Of note is that among the three favorable energy terms, Δ*E*_vdW_ alone can overcompensate for the considerable negative contribution of Δ*G*_polar_ and, hence, is the dominant factor in determining the high-affinity binding of the four inhibitors to Mpro.

Interestingly, the ranking of the Mpro-binding affinity of the four inhibitors is consistent with that of the inter-molecular vdW interaction strength. However, the differences in values of the other energy terms can also contribute to the differences in the Mpro-binding affinity among the four inhibitors. For example, when compared to ML188, ECI2 has only a slightly stronger vdW interaction with Mpro (−3.1 kJ/mol), which cannot compensate for the negative effects from its much weaker electrostatic interaction with Mpro (21.3 kJ/mol) and from the slightly less favorable non-polar solvation contribution (1.8 kJ/mol). Nevertheless, the much lower electrostatic desolvation penalty (−27.4 kJ/mol) overcompensates for the total negative effect (23.1 kJ/mol), thus contributing substantially to the increased Mpro-binding affinity of ECI2 compared to ML188 (−7.4 kJ/mol). As a matter of fact, the weakest binding affinity of Mpro-ECI3 (−83.0 kJ/mol) stems primarily from its highest electrostatic desolvation penalty among the four complexes, although it has the strongest electrostatic interaction strength and the weakest vdW interaction strength.

Figure 7 shows the per-residue BFE average values of Mpro within the four complexes. It is clear that in all the four inhibitor-bound Mpros, (i) only a few residues have BFE values greater than 3.0 kJ/mol (primary negative contributing residues (PNCRs) to the inhibitor-binding affinity) or lower than −3.0 kJ/mol (primary positive contributing residues (PPCRs)), and (ii) the number of PPCRs is far higher than that of PNCRs. All these PPCRs/PNCRs participate in the formation of the inhibitor/substrate-binding cavity of Mpro (Figure 5A), although their contribution magnitudes differ among the four Mpro forms. Notably, Glu166 is the most prominent negative contributing residue in the ML188-, ECI3-, and ECI4-bound Mpro, with the largest negative contribution observed in the ECI3-bound Mpro. In contrast, Glu166 makes a slightly positive contribution to the ECI2-binding affinity. Since Mpro-ECI3 and Mpro-ECI2 have the highest and lowest Δ*G*_binding_ values (Table 2), respectively, the above observations indicate that Glu166 plays a crucial role in influencing the inhibitor-binding affinity.

Among the PPCRs, Met165 not only contributes most significantly to the ML188- and ECI4-binding affinities, but also considerably to the ECI2- and ECI3-binding affinities. Met49 and Cys145 contribute to enhancing the Mpro affinity to three inhibitors. His41, Pro168, His172, Asp187, and Gln189 contribute to enhancing the Mpro affinity to two inhibitors. Leu141 and Leu167 are inhibitor-specific affinity-enhancing residues. Given the above observations, it can be considered that Met165 is the most crucial affinity-enhancing residue in terms of both the inhibitor’s breadth and enhancing ability. Nevertheless, the other identified PPCRs are still important target residues for further improving the binding affinity via ligand modification.

### 2.8. Binding Mode Analysis

For each complex, the frame/snapshot with the BFE median was extracted from the representative structural ensemble and used for the protein-ligand binding mode/interaction analysis. Figure 8 shows the 2D binding modes of the four inhibitors to the substrate-binding cavity of Mpro. For ML188, its three rings (furan, pyridine, and benzene) interact mainly with residues from the pockets S1′, S1, and S2, respectively (Figure 8A). However, the interactions of ML188 with the S4 pocket are rare. ECI2 interacts mainly with residues from the pockets S1′, S2, and S4, but rarely with residues from the S1 pocket (Figure 8B). For ECI3, its different rings/groups mainly interact with residues from the pockets S1′, S1, and S4, but rarely with residues from the S2 pocket (Figure 8C). For ECI4, its chlorobenzene ring interacts intensively with the S1 pocket residues while the other rings/groups interact mainly with residues from the pockets S2 and S4 (Figure 8D).

Overall, the numbers of Mpro residues involved in the direct interactions with inhibitors differ slightly among the four complexes. It is apparent that the vdW contacts dominate over the other interactions in all the four complexes. Note that the alkyl-related interactions, which originate from the London dispersion forces and thus belong to the vdW category [65], also occur in relatively high numbers in the four complexes. These explain the earlier observations that the inter-molecular vdW interaction strength is much stronger than the inter-molecular electrostatic strength in all the four complexes (Figure 5C,D, Table 2).

Due to the lack of a charged group in all the four inhibitors, no ionic pair interaction was observed between inhibitors and Mpro. However, the other electrostatic-originated interactions, including the conventional HBs, weak HBs, and π-related (i.e., Pi-related stacked, Pi-sulfur, Pi-anion, and Pi-lone pair interactions) interactions, were somewhat observed. Since the conventional HBs are more than two-fold stronger than the weak HBs and π-related interactions, the presence of multiple conventional HBs in Mpro-ML188 (four; Figure 8A) and Mpro-ECI3 (three; Figure 8C) likely explain their lower inter-molecular average electrostatic energies compared to Mpro-ECI2 and Mpro-ECI4 (Figure 5D and Table 2), which have only few weak HBs and π-related interactions in the respective representative snapshots (Figure 8B,D).

Notably, Met165 (from the S4 pocket) simultaneously forms π-related stacked, π-sulfur, vdW, and alkyl-related interactions with different moieties of ECI4 (Figure 8D). This, in conjunction with a small buried surface area of Met165, explains its largest positive contribution to the ECI4-binding affinity of Mpro (Figure 7). Additionally, worth noting is that although Glu166 (from the S1 pocket) forms one conventional HB with ML188, two conventional HBs and one π-anion interaction with ECI3, and two weak HBs and one π-lone pair interaction with ECI4, it makes the largest negative contribution to the binding affinity of Mpro to the three inhibitors (Figure 7). For each of the above three complexes, Glu166 has the largest buried surface area among all the interface residues (Figure 8A,C,D). This results in the electrostatic desolvation penalty that is high enough to counteract the favorable enthalpic contributions.

In summary, despite the different binding modes of the four inhibitors to Mpro, they all interact with Mpro via the preponderant vdW interactions and minor electrostatic-originated interactions. Although the electrostatic-originated interactions are often thought to play an important role in enhancing the inter-molecular interaction strength, they are not necessarily able to offset the negative effect of the electrostatic desolvation on the binding affinity. In future ligand modification, the inter-molecular HBs involving Glu166 should be avoided due to its high desolvation cost paid for the HB formation.

## 3. Discussion

In order to obtain potential non-covalent drug compounds with inhibitory effects on the SARS-CoV-2 Mpro, we performed a comprehensive study using a combined computational strategy. Specifically, initial pharmacophore-based VS of the ZINC purchasable database yielded 593 hit compounds, which were further screened by molecular docking, obtaining 78 potential candidate compounds with a “docking” affinity no less than that of the reference inhibitor ML188. The subsequent filtering with the predicted ADMET parameters yielded four ECIs (ECI1-4) exhibiting no potential toxicities but satisfactory pharmacokinetic and drug-like properties.

Further MD simulations revealed that ML188 and ECI2-4 kept staying within the substrate-binding cavity of Mpro during the entire simulation period, while ECI1 (which has the lowest/best docking score value among the five inhibitors; see Appendix A) escaped out of the cavity. This is not surprising, as the optimal docking pose of a ligand (i.e., the pose with the best docking score) does not necessarily represent the actual binding poses within the protein target, whose space could be sampled by the all-atom MD simulation with a more sophisticated force field under explicit conditions (e.g., solvent, pH, salt concentration, temperature, and pressure) [66,67,68,69]. Our results highlight that the MD simulation following the molecular docking is necessary for assessing the complex stability and optimizing the inter-molecular binding mode.

Inter-molecular vdW and electrostatic interactions are the predominant forces maintaining the stable association between the ligand and receptor [68]. For the reference complex (Mpro-ML188) and the three effective complexes (Mpro-ECI2, Mpro-ECI3, and Mpro-ECI4), the inter-molecular interaction potential energies calculated both from the equilibrated MD trajectories (Figure 5C,D) and the representative structural ensembles (Table 2) reveal that the vdW forces are far stronger than the electrostatic forces and, hence, play a substantial role in determining the steady placement of these inhibitors within the substrate-binding cavity. The comparison between MM-PBSA energy components of different complexes further reveals that the inter-molecular vdW interactions are the dominant forces in determining the high binding affinities of these inhibitors to Mpro (Table 2). This is corroborated by the overwhelming number of the inter-molecular vdW contacts over that of the electrostatic-originated interactions observed in the representative structures of individual complexes (Figure 8). Furthermore, our findings agree with previous studies [70,71] suggesting that enhancing the inter-molecular vdW forces is the main direction for improving the binding affinity of a lead compound to Mpro.

Interestingly, we observed a rough trend that as the dispersion of the vdW energy distribution increases (Figure 5C), the inhibitor mobility (Figure 4), the protein interface flexibility (Figure 5A), and the interface entropy (Figure 6) also increase. This indicates that a decrease in stabilization of the inter-molecular vdW forces is likely to be responsible for the increased instability of the bound state.

Among the various types of electrostatic-originated interactions, the conventional HB is generally thought to be very important in stabilizing the structure of a protein or a protein-ligand complex due to its high strength and relatively high number [68,72]. In the case of the four complexes studied here, the features of the inter-molecular HB number distribution (Figure 5B) conform to those of the inter-molecular electrostatic interaction energy distribution (Figure 5D), indicating that the HB dominates over the other electrostatic-originated interactions, with its number and number variability dictating the strength and variability of the inter-molecular electrostatic forces.

It should be noted that the strengths of the individual inter-molecular HBs are different, depending on the electronegativity of the hydrogen bond donor (HBD) and HBA, the linearity of HBD, hydrogen and HBA, and the surrounding microscopic environment. The differences in the inter-molecular HB number distribution among the four complexes likely reflect the HB strength differences. Among the four complexes, Mpro-ML188 has the highest percentage of frames carrying three or more inter-molecular HBs, while Mpro-ECI3 shows the most dispersed HB number distribution (Figure 5B). Therefore, it can be considered that the inter-molecular HBs are generally stronger in ML188-Mpro than in ECI3-Mpro. During the MD simulation, the low-strength HBs are more readily broken and have shorter lifetimes than the high-strength HBs. Nevertheless, the broken HB can be reformed, either between the original HBD and HBA or between the original HBD/HBA and the new HBA/HBD. We defined the process of HB breaking and reformation between alternative HBD and HBA as the competitive HB interactions, which were shown to play an important role in facilitating protein conformational fluctuations in our previous study [73].

For Mpro-ML188, the multiple stable but less competitive inter-molecular HB interactions cooperate with the most stable and strongest inter-molecular vdW interactions, ultimately resulting in the smallest interface entropy (Figure 6A) and the least ML188 displacement within the substrate-binding cavity (Figure 4A). For Mpro-ECI3, the intense inter-molecular competitive HB interactions, in conjunction with its weakest inter-molecular average vdW force (Figure 5C), explain the largest interface entropy (Figure 6C) and the most dramatic displacement of ECI3 within the substrate cavity (Figure 4D). For the two complexes (Mpro-ECI2 and Mpro-ECI4) with fewer inter-molecular HBs, the inter-molecular HB competition is slightly more intense in Mpro-ECI2 due to its more dispersed HB number distribution (Figure 5B). This, together with the higher variability of the inter-molecular vdW forces in Mpro-ECI2 (Figure 5C), explains its slightly increased interface entropy and ligand displacement compared to Mpro-ECI4.

Comparisons between the MM-PBSA energy components of the four complexes reveal that the differences in the electrostatic desolvation free energy (Δ*G*_polar_) determine the differences in the total BFE (Δ*G*_binding_). We also found an interesting phenomenon that a small increase in the inter-molecular electrostatic interaction strength can lead to a large increase in the electrostatic desolvation energy penalty (Table 2), which over-offsets the positive effect of the increased electrostatic enthalpic contribution and ultimately impairs the binding affinity. As a matter of fact, the electrostatic desolvation free energy is the work performed to break the favorable electrostatic interactions between the solvent (water) and solute (protein and ligand) when displacing water molecules from the binding interfaces. Upon the formation of a complex, the larger the buried polar surface areas, the more work is required to strip off the hydration water around the polar groups/atoms in the unbound states. Although the favorable electrostatic interactions (e.g., ionic pair or HB) may be formed between the buried polar/charged protein residues and ligand moieties and, hence, contribute to lowering BFE, such a reduction in the electrostatic enthalpy does not necessarily counteract the negative effect of the electrostatic desolvation penalty. A typical example is the Glu166, which although forms favorable electrostatic interactions with ML188, ECI3, and ECI4 (Figure 8A,C,D), makes a significant negative contribution to the binding affinity of Mpro to the three inhibitors (Figure 7) due to its high degree of burial (Figure 8). In the case of Mpro-ECI2, Glu166 forms neither vdW contacts nor direct electrostatic interactions with ECI2 (Figure 5A). This implies that the complex formation has no impact on the solvent-exposure extent of Glu166, which results in a negligible desolvation penalty and hence, explains its slightly favorable contribution to the total BFE (Figure 7; ECI2-bound Mpro). As a matter of fact, Glu166 participates in the formation of the S1 pocket (Figure 1D). Although the methylbenzene moiety of ECI2 makes contacts with the S1′/S1 bordering residues (i.e., Asn142, Cys145, and His164; see Figure 8B), this moiety does not cross the border and occupy the S1 pocket.

The common features for the identified PPCRs (Figure 7) are that (i) they interact with different inhibitors through multiple vdW contacts and the electrostatic-originated interactions that involve almost no conventional HB and, (ii) their burial degrees upon binding are limited (Figure 8). These features indicate that (i) multiple vdW contacts and electrostatic-originated non-HB interactions formed between a Mpro interface residue and the inhibitor can accumulate sufficiently strong enthalpic forces capable of contributing substantially to lowering the residue BFE, and (ii) the negative effect of the desolvation energy penalty can be overcompensated for by the cumulative favorable enthalpic contributions if the burial degree of the interface residue is limited.

Although all the four pockets (S1′, S1, S2, and S4) contain the mixed polar and non-polar residues (Figure 1D), they exhibit different degrees of hydrophobicity: S1′ > S1 > S4 > S2 (Figure 1C). However, it seems that the inhibitor-affinity contributions of the identified PPCRs and PNCRs are irrelevant to the overall hydrophobicity of a pocket, but depend on their locations in the pocket, which in turn determine their burial degrees upon binding. For example, Met49 is located at the outer rim of the hydrophilic S2 pocket (Figure 1C,D). The formation of the favorable vdW/electrostatic-originated interactions between Met49 and the benzene rings of ML188, ECI2, and ECI4 limitedly affects the burial degree of Met49 (Figure 8A,B,D), thus, explaining its significant contribution to enhancing the binding affinity of Mpro to the three inhibitors. The side chain of Glu166 lines one side wall and the partial bottom of the S1 pocket that has a relatively high overall hydrophobicity. Therefore, the full occupancy of S1 by the inhibitor moiety will inevitably bury the polar surface of Glu166 and displace the surrounding hydration water, thus, explaining its high desolvation penalty and significant negative contribution to ML188-, ECI3-, and ECI4-binding affinities. In addition, we notice that, despite the high hydrophobicity of the S1′ pocket, it contains almost no PPCR residues, with the exception of His41 and Cys145. These two residues are catalytic dyad residues that border between S1′ and S2 and between S1′ and S1, respectively.

To this end, we suggest that the future modifications dedicated to improving the Mpro-affinity of the three ECIs may follow the following three principles: (i) increasing the number of vdW contacts and/or electrostatic-originated non-HB interactions with the substrate-binding cavity (in particular with S1′) of Mpro through the introduction of non-polar and/or aromatic groups, (ii) removing/replacing the polar groups/atoms that form HBs with the “easy-to-be-buried” Glu166, and (iii) introducing the strong HBs under the premise of limited burials of the polar surface areas of the involved Mpro residues and inhibitor moieties upon binding.

## 4. Materials and Methods

### 4.1. Pharmacophore Model Generation and Initial Screening

The crystal structure of the SARS-CoV-2 Mpro-ML188 complex was obtained from the Protein Data Bank (PDB; http://www.rcsb.org (accessed on 18 July 2022)) with PDB ID 7L0D [34]. The crystallographic water molecules and heteroatoms were removed, and the atomic coordinates of Mpro and ML188 were saved as separate PDB files, which were used as the receptor and ligand inputs, respectively, for the Pharmit [74] online server (http://pharmit.csb.pitt.edu (accessed on 20 July 2022)) to generate the pharmacophore model. The pharmacophore features need to satisfy the following criteria: the distance of the HBD/HBA in the ligand is within 4 Å of the HBA/HBD in the receptor; the charged group in the ligand is within 5 Å from the oppositely charged group in the receptor; the distance between centroids of the aromatic groups from the ligand and receptor is within 5 Å; the hydrophobic group in the ligand is within 6 Å from at least three hydrophobic groups in the receptor. Apart from specifying pharmacophore features, the Ro5 [75] was used as a filter to screen compounds with acceptable drug-likeness properties. Finally, the initial screening was performed on more than 13,190,317 compounds in the ZINC purchasable database (a pre-built library in Pharmit) using Pharmit with the generated pharmacophore model.

### 4.2. Preparation of the SARS-CoV-2 Mpro Structure

The unresolved C-terminal residue Gln306 in the crystal structure 7L0D was modeled using PyMOL [76], and the missing side-chain atomic coordinates of Thr45, Glu47, Asp48, Arg60, Arg76, Lys100, Lys102, Lys137, and Arg222 were built using the H++ [77] online server (http://newbiophysics.cs.vt.edu/H++/ (accessed on 8 August 2022)). The missing hydrogen atoms in the Mpro structure were also added by H++, with the protonation states of all titratable residues assigned according to their predicted pKa values at pH 7.4. The complete Mpro structure obtained will be used for molecular docking.

### 4.3. Molecular Docking

The hit compounds (ligands) from the initial screening were subjected to further screening by molecular docking to Mpro (receptor) using AutoDock Vina 1.2.0 [78]. ML188 was also docked to Mpro to obtain the scoring value, which was used as a reference for comparison with the docking values obtained for the hit compounds. For the Mpro structure, AutoDockTools 1.5.6 [79] was used to merge the non-polar hydrogen atoms, assign Gasterier partial charges [80], and convert the PDB file to the PDBQT format suitable for calculating energy grid maps. For the ligand molecules, Open Babel 3.0.0 [81] was used to convert molecules from the SDF to PDBQT format, add hydrogens, assign protonation states of the titratable groups (at pH 7.4), and assign Gasteiger partial charges.

A box size of 24 × 24 × 24 Å with a grid spacing of 1.0 Å was considered the search space region in Mpro, with the box center defined as the center of the co-crystallized ML188. In docking, the Mpro structure was kept rigid while the ligand molecules were left fully flexible; the AutoDock Vina default parameters (e.g., exhaustiveness = 8, energy_range = 3.0 kcal/mol, and sf_name = ‘vina’) were used with the exception of the number of poses to generate (n_poses), which was set to 10.

The validity of the docking results was evaluated by comparing the binding mode of the co-crystallized ML188 against Mpro (PDB ID: 7L0D) [34] with that of the docked ML188. The RMSD cut-off of 1.5 Å was used as a docking validity criterion. Ligands with the best scoring values equal to or lower than that of ML188 were considered PCCs and were subjected to predictions of pharmacokinetic properties.

### 4.4. ADMET Prediction

The pkCSM [82] online server (http://biosig.unimelb.edu.au/pkcsm (accessed on 14 August 2022)) was used to predict the pharmacokinetic properties/ADMET parameters (absorption, distribution, metabolism, excretion, and toxicity) [83] of PCCs and ML188. The simplified molecular-input line-entry specification (SMILES) strings of PCCs and ML188, which are the input format required by pkCSM, were obtained from the ZINC15 [84] and MedCHemExpress (http://www.medchemexpress.com (accessed on 17 August 2022)) databases, respectively. Those compounds, with satisfactory ADMET profiles, were considered as ECIs, for which the Mpro-ligand complex structures were subjected to MD simulations to validate the structural stability and investigate the nature and strength of inter-molecular interactions.

### 4.5. Molecular Dynamics Simulation

All MD simulations were performed using the GROMACS 5.1.4 package [85] with the Amber ff99SB force field (FF) [86]. The ligand FF parameters were generated by the general Amber force field (GAFF) using the AmberTools20 package [87], and the atomic partial charges were calculated using the Austin Model 1-bond charge corrections (AM1-BCC) method [88] in GAFF.

Each protein-ligand complex, which retains the earlier assigned protonation states of the titratable residues and ligand groups, was placed in a periodic dodecahedron box of TIP3P water [89] with the minimum solute-box wall distance of 12 Å. Enough Na^+^ and Cl^−^ ions were added to neutralize the system and mimic the physiological salt concentration of 0.15 M. Each prepared system was subjected to the steepest descent, followed by the conjugate gradient energy minimization, until the convergence criterion of 10 kJ/mol/nm was satisfied. This was followed by four consecutive NVT MD runs (each performed for 300 ps at 300 K with a time step of 1 fs) with position restraints on the solute heavy atoms by decreasing the harmonic potential force constant (1000, 100, 10, and 0 kJ/mol/nm^2^) to equilibrate the solvent around the solute.

Finally, each system was subjected to a 100-ns production MD simulation with the following protocols/parameters: All bond lengths were constrained using the LINear constraint solver (LINCS) algorithm [90], allowing a time step of 2 fs. The Verlet cut-off scheme was used for the neighbor list with a 0.005 kJ/mol/ps buffer tolerance [91]. The particle mesh Ewald (PME) method [92] was used to calculate the long-range electrostatic interactions, with a real-space cut-off of 1.2 nm, grid spacing of 0.12 nm, and an interpolation order of 4. The vdW interactions were calculated using the Lennard-Jones (LJ) potential truncated at a distance of 1.2 nm. The system temperature was coupled using the v-rescaling algorithm [93] at 300 K with a time constant of 0.1 ps, whereas the pressure was coupled to 1 bar using the Parrinello–Rahman algorithm [94] with a time constant of 2.0 ps. The atomic coordinates were saved every 2 ps.

### 4.6. Post-Dynamics Analysis

For the MD trajectory of each complex, the RMSD values of the protein (Mpro) C_α_ atoms and the ligand heavy atoms with respect to the starting structure were calculated using the GROMACS tool ‘gmx rms’. The structural stability of the complex and the equilibrated portion of a trajectory were assessed based on the time-dependent RMSD profile. To avoid artifacts arising from the simulation procedure, all the other geometric and energetic parameters were calculated from the equilibrated portion of each trajectory. Specifically, the RMSF, inter-atomic distance between the ligand and protein, solvent accessible surface area (SASA), inter-molecular HBs, and inter-molecular interaction potential energies (i.e., vdW and electrostatic energies) were computed using the GROMACS tools ‘gmx rmsf’, ‘gmx mindist’, ‘gmx sasa’, ‘gmx hbond’, and ‘gmx energy’, respectively.

The protein residues with the average minimum distance of ≤ 4 Å to the ligand were considered the binding interface residues. The buried surface area was obtained by subtracting the complex SASA value from the sum of the SASA values of the ligand and protein. An inter-molecular HB was considered to exist when the donor-acceptor distance is less than 3.5 Å and the donor-hydrogen-acceptor angle is greater than 120°.

### 4.7. FEL Reconstruction

The two-dimensional FEL was reconstructed using the iRMSD (which was calculated on the heavy atoms of the protein interface residues and the ligand) and buried surface area as the reaction coordinates with the following equation:(1)$$F\left(s\right)=-kT\ln\left(\frac{P_{i}}{P_{\max}}\right)$$
where *k* and *T* are the Boltzmann’s constant and simulation temperature, respectively. $P_{i}$ is the probability of finding the system in state i, characterized by the selected reaction coordinates, and $P_{\max}$ is the probability of the most probable state. An in-house Python script was used to generate FELs of the protein-ligand complexes (Appendix A). For each complex, 500 conformations were randomly extracted from the global-free energy minimum of FEL and were treated as the representative structural ensemble for the subsequent BFE calculation.

### 4.8. BFE Calculation

The molecular mechanics Possion–Boltzmann surface area (MM-PBSA) method implemented in the GROMACS tool g_mmpbsa [95] was used to calculate the BFE (Δ*G*_binding_) of each protein-ligand complex as follows:Δ*G*_binding_ = Δ*E*_MM_ + Δ*G*_solvation_ − *T*Δ*S =* (Δ*E*_bonded_ *+* Δ*E*_vdW_ + Δ*E*_elec_) + (Δ*G*_polar_ + Δ*G*_non-polar_) − *T*Δ*S*(2)
where Δ*E*_MM_, Δ*G*_solvation_, and *T*Δ*S* are the changes upon binding in the vacuum molecular mechanics potential energy, solvation free energy, and solute entropy, respectively. Δ*E*_MM_ can be decomposed into the changes in the bonded (Δ*E*_bonded_), vdW (Δ*E*_vdW_), and electrostatic (Δ*E*_elec_) energies, whereas Δ*G*_solvation_ into the contributions of the polar (Δ*G*_polar_) and non-polar (Δ*G*_non-polar_) solvation free energies. Note that the Δ*E*_bonded_ value is zero due to the single trajectory approach used here. The *T*Δ*S* term was not included in our calculations due to the high uncertainty of the estimated results and its minimal effect on comparing the binding affinities of different ligands to the same receptor when it is omitted.

In the MM-PBSA calculation, the g_mmpbsa default parameters were used with the following exceptions in estimating Δ*G*_polar_: the grid resolution of 0.5 Å, the ionic strength of 0.15 M, and the dielectric constants of the solute and solvent of 4 and 80, respectively.

Residue BFE (i.e., per-residue contribution to the total BFE) was calculated using the ‘binding energy decomposition’ module of g_mmpbsa, which decomposes the total BFE into contributions from individual protein residues by calculating each atom’s energy components (i.e., *E*_MM_, *G*_polar_, and *G*_non-polar_) in a residue in both the free and bound forms.

### 4.9. Binding Mode Analysis

For each complex, the frame/snapshot with the median BFE was extracted from the structural ensemble and was considered the representative structure. For each representative structure, the binding mode of protein-ligand interactions was analyzed using the Biovia Discovery Studio 2020 Client [96].

## 5. Conclusions

In this work, we first screened from the ZINC purchasable database the three potential non-covalent drug compounds against the SARS-CoV-2 Mpro using a computational framework that combines the techniques of structure-based pharmacophore modeling, pharmacophore-based VS, molecular docking, pharmacokinetic parameter prediction, and MD simulation. The subsequent comparative analyses of the four complexes (three effective complexes and the reference complex Mpro-Ml188) in terms of dynamics, thermodynamics, BFE, and inter-molecular interactions not only reveal the differences in stability, binding affinity, and inter-molecular interaction forces and modes among them, but also elucidate the mechanical and energetic mechanisms underlying these differences. Our main findings are as follows: First, the inter-molecular vdW forces are far more important than the inter-molecular electrostatic forces in maintaining the stable association of the four inhibitors with Mpro. Second, the inter-molecular competitive HB interactions are not conducive to the stable association of an inhibitor with Mpro. Third, the inter-molecular vdW interaction energy is the determinant of the high binding affinities of the four inhibitors to Mpro. Fourth, the electrostatic desolvation energy penalty is the determinant of different Mpro-binding affinities of the four inhibitors. Based on these findings, we conclude that enhancing the inter-molecular vdW interactions while avoiding introducing polar groups/atoms capable of causing a large electrostatic desolvation penalty upon HB formation may be a promising strategy in future inhibitor optimization and drug lead design.

## Figures and Tables

**Figure 1 ijms-24-04237-f001:**
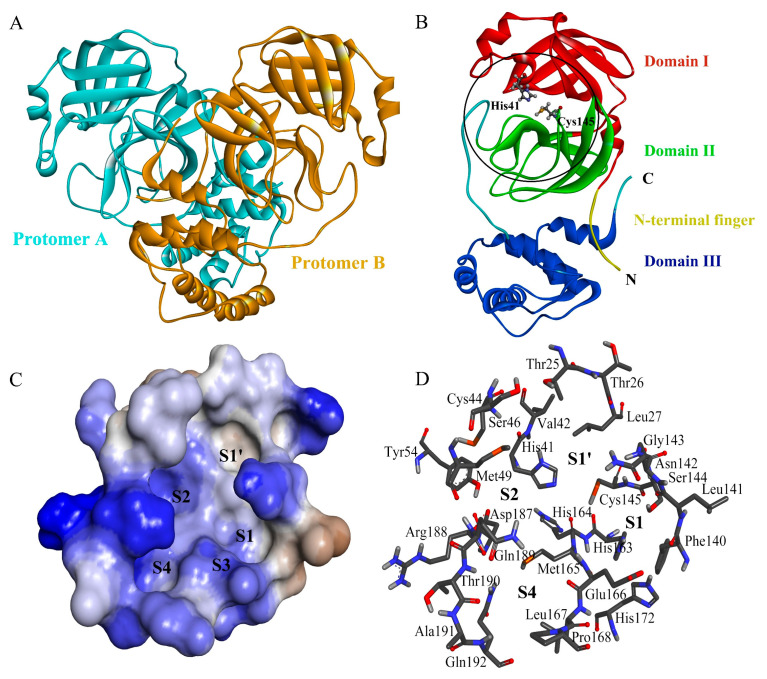
Three-dimensional structure of the SARS-CoV-2 Mpro and its substrate-binding subsite/pockets. (**A**) Cartoon representation of the crystal structure of the Mpro dimer (PDB ID: 7ALI [30]). The two protomers/monomers are colored cyan and bronze, respectively; (**B**) cartoon representation of the crystal structure of an Mpro protomer (PDB ID: 7L0D [34]). The N-terminal finger and domains I–III are colored differently. The catalytic dyad residues, Cys145 and His41, are shown in ball-and-stick representation; (**C**) surface representation of Mpro showing the substrate-binding pockets/subsites S4, S3, S2, S1, and S1′. The protein surface is colored by hydrophobicity, with the color gradient ranging from blue for the most hydrophilic surface to brown for the most hydrophobic surface; (**D**) amino acid residues that form the substrate-binding pockets S4, S2, S1, and S1′. Residues are shown in stick representation, with C, O, N, S, and H atoms colored in gray, red, blue, orange, and white, respectively.

**Figure 2 ijms-24-04237-f002:**
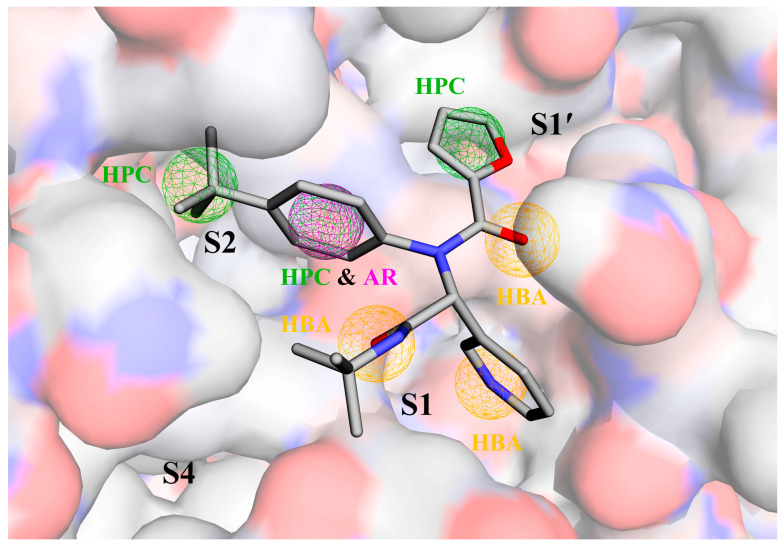
The pharmacophore model generated by the Pharmit server in the substrate-binding cavity of the SARS-CoV-2 Mpro (PDB ID: 7L0D [34]). Mpro and ML188 are shown as surface and stick representations, respectively, with the red, blue, and gray regions/atoms corresponding to O, N, and C atoms, respectively. Pharmacophore features are represented by mesh spheres with the following color codes: orange for the hydrogen bond acceptor (HBA), green for the hydrophobic center (HPC), and purple for the aromatic ring (AR).

**Figure 3 ijms-24-04237-f003:**
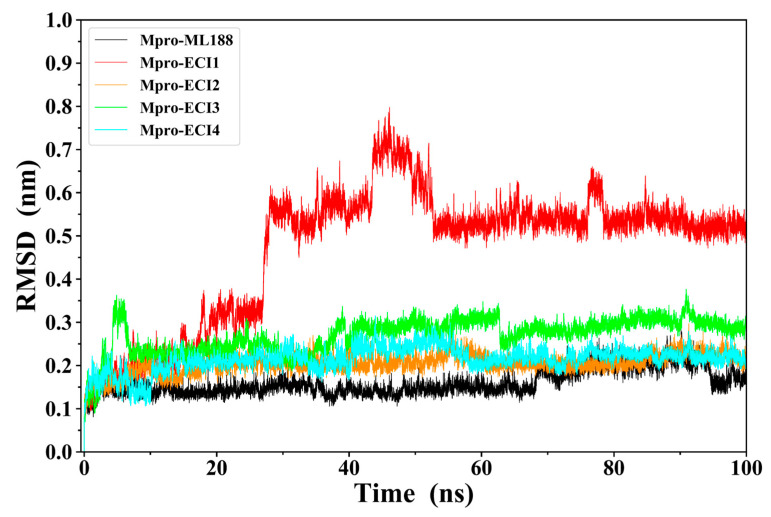
Time-dependent root mean square deviation (RMSD) values of Mpro C_α_ and inhibitor heavy atoms of the five complexes with respect to their respective starting structures during the 100-ns MD simulation.

**Figure 4 ijms-24-04237-f004:**
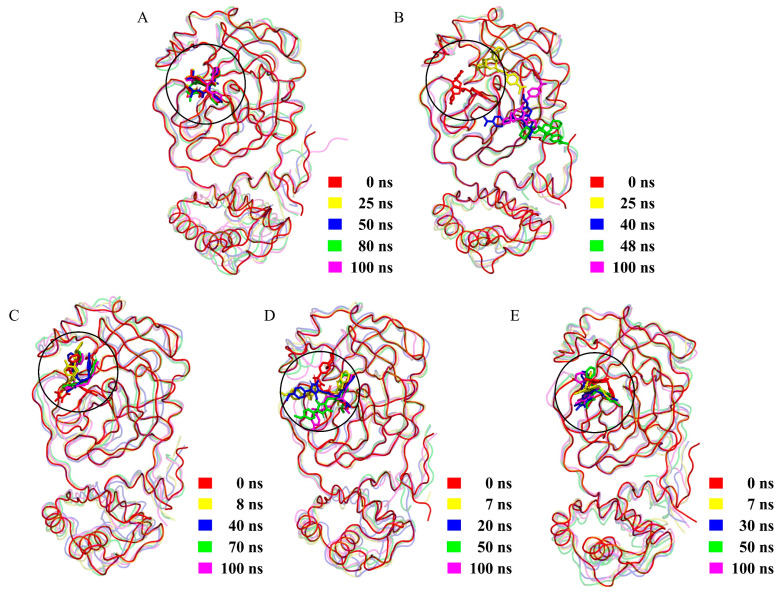
Structural superposition obtained by least-square fitting of the four snapshots extracted from the MD simulation trajectory at different times to the starting structure. (**A**) Mpro-ML188. (**B**) Mpro-ECI1. (**C**) Mpro-ECI2. (**D**) Mpro-ECI3. (**E**) Mpro-ECI4. Mpro and ML188/ECIs are shown in tube and stick representations, respectively, with the substrate-binding site highlighted by a circle. For convenience of observation, in all complexes, the Mpro structure at 0 ns (starting structure) is represented as an opaque tube while those at the other times as a semi-transparent tube.

**Figure 5 ijms-24-04237-f005:**
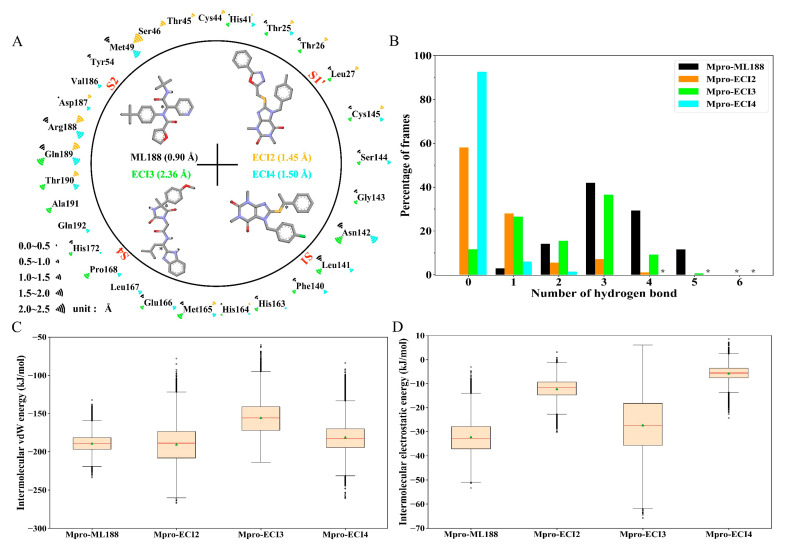
Interface residues and inter-molecular interaction statistics of Mpro-ML188 and Mpro-ECI2-4 during the equilibration simulations. (**A**) Identified Mpro interface residues for the four inhibitors. Skeletal formulas of ML188 and ECI2-4, along with their full-atom root mean square fluctuation (RMSF) values (in parentheses), are shown within the circle. The chiral centers in these molecules are indicated by an asterisk. The magnitudes of the all-atom RMSF values of the Mpro interface residues are marked using the Wifi-like signal strength icon with different colors. (**B**) Histogram showing the distributions of the inter-molecular hydrogen bond (HB) number. (**C**) Boxplot showing the inter-molecular van der Waals (vdW) interaction energy distributions. (**D**) Boxplot showing the inter-molecular electrostatic interaction energy distributions.

**Figure 6 ijms-24-04237-f006:**
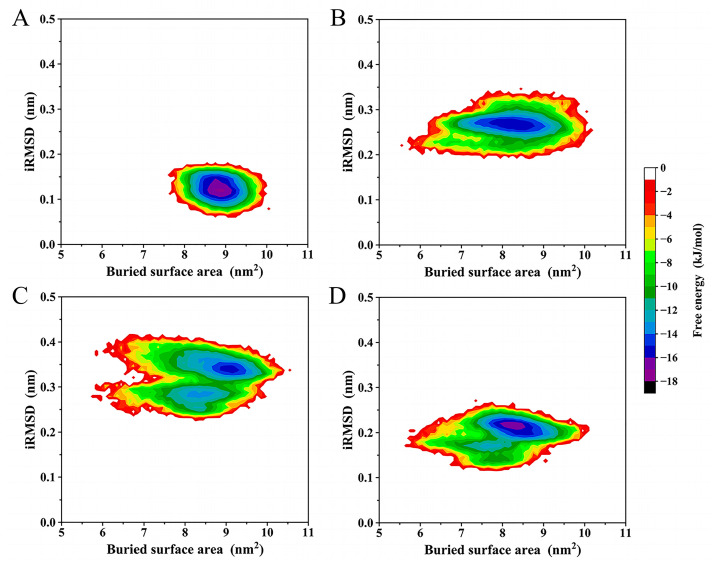
Free energy landscapes (FELs) constructed using the buried surface area and interface RMSD (iRMSD) as the reaction coordinates. (**A**–**D**) FELs of Mpro-ML188, Mpro-ECI2, Mpro-ECI3, and Mpro-ECI4, respectively. The color bar denotes the relative free energy value in kJ/mol.

**Figure 7 ijms-24-04237-f007:**
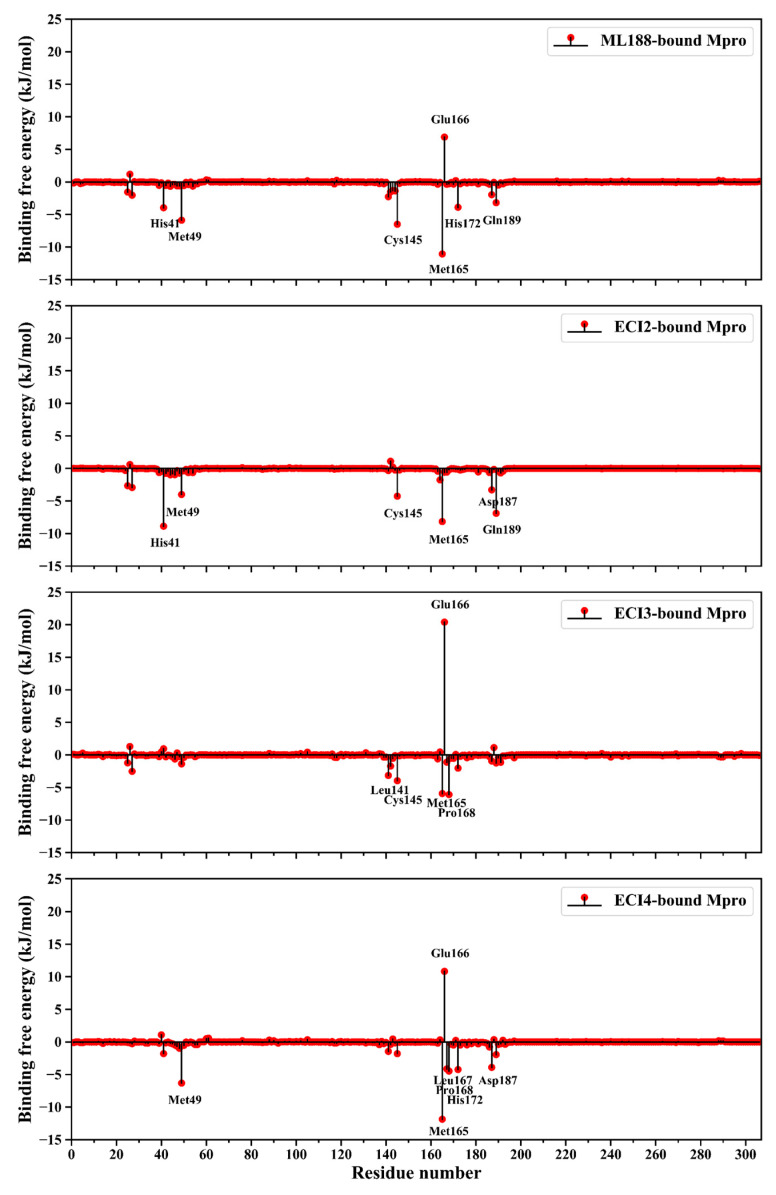
Per-residue binding free energy (BFE) decompositions of ML188-, ECI2-, ECI3-, and ECI4-bound Mpro. Residues with BFE values lower than −3.0 kJ/mol (primary positive contributing residues (PPCRs)) or greater than 3.0 kJ/mol (primary negative contributing residues (PNCRs)) are labeled.

**Figure 8 ijms-24-04237-f008:**
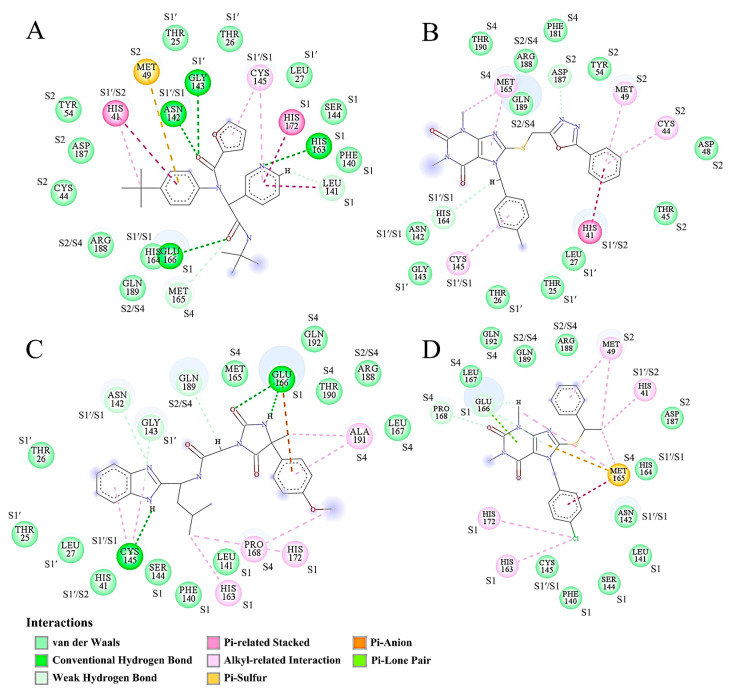
Binding modes of the representative structures of the four Mpro-inhibitor complexes. The binding mode is shown by the 2D interaction diagram of ML188 (**A**), ECI2 (**B**), ECI3 (**C**), and ECI4 (**D**) with Mpro residues. Dashed lines with different colors represent different interaction types other than vdW contacts. Each circle with the same size represents an Mpro residue and is colored according to the main interaction type involved. The size of the light-gray circle surrounding the residue circle varies proportionally to the amount of the buried surface area upon binding. The pocket label beside each Mpro residue indicates which pocket(s) the residue belongs to. The residue with a double label (e.g., S1′/S1) indicates that it participates in the formation of two different pockets or borders between the two pockets.

**Table 1 ijms-24-04237-t001:** Predicted parameters of Lipinski’s rule of five (Ro5) and ADMET for the four effective candidate inhibitors (ECI1-4) and ML188.

Property	Parameter	ECI1	ECI2	ECI3	ECI4	ML188
Ro5	MW ^a^	472.53	474.55	463.54	440.96	433.55
	LogP ^b^	4.04	3.13	3.24	3.99	5.27
	HBD ^c^	0	0	3	0	1
	HBA ^d^	8	10	5	7	4
Absorption	Water solubility ^e^	−4.17	−2.92	−2.95	−3.45	−4.76
	Intestinal absorption ^f^	97.3%	93.8%	73.8%	95.9%	95.0%
Distribution	VDss ^g^	0.24	−0.29	0.02	0.53	0.52
	BBB permeability ^h^	−0.96	−1.37	−0.69	0.41	−0.11
Metabolism	CYP2D6 substrate ^i^	No	No	No	No	No
	CYP3A4 substrate ^j^	Yes	Yes	Yes	Yes	Yes
	CYP2D6 inhibitor ^i^	No	No	No	No	No
	CYP3A4 inhibitor ^j^	Yes	Yes	Yes	Yes	Yes
Excretion	Renal OCT2 substrate ^k^	No	Yes	No	Yes	No
	Total clearance ^l^	0.62	0.21	0.74	0.24	0.50
Toxicity	AMES toxicity ^m^	No	No	No	No	Yes
	Rat LD50 ^n^	2.95	2.57	2.39	2.89	3.23
	Hepatotoxicity ^o^	No	No	No	No	Yes

^a^ Molecular weight in Daltons. ^b^ Octanol-water partition coefficient (logP). ^c^ Number of hydrogen bond donor (HBD). ^d^ Number of hydrogen bond acceptor (HBA). ^e^ Water solubility is given as the logarithm of the molar concentration (log(mol/L)). ^f^ The percentage that will be absorbed through the human intestine. ^g^ Volume of distribution at steady state (VDss) is given as log(L/kg), with the value < −0.15 and > 0.45 considered to be more inclined to be distributed in the blood plasma and tissue, respectively. ^h^ Blood brain-barrier (BBB) is given as the logarithmic ratio of brain to plasma drug concentrations (logBB), with the value >0.3 and <−1.0 considered to readily and poorly cross the BBB, respectively. ^i^ Whether a given compound is likely to be the substrate or inhibitor of the main isoform of cytochrome P450 (CYP) enzymes, CYP2D6. ^j^ Whether a compound is likely to be the substrate or inhibitor of another main CYP isoform, CYP3A4. ^k^ Whether a compound is likely to be the substrate of renal organic cation transporter 2 (OCT2). ^l^ Total clearance is given as the logarithm of the proportionality constant CLtot (log(ml/min/kg)). ^m^ Whether a compound is likely to be mutagenic/carcinogenic. ^n^ Rat median lethal dose (LD50) in the unit of mol/kg for a measurement of acute toxicity. ^o^ Whether a compound is likely to cause liver injury.

**Table 2 ijms-24-04237-t002:** Average values and corresponding standard deviations (in parentheses) of the MM-PBSA binding free energy (BFE) and energy components (kJ/mol) calculated from the respective structural ensembles of the four Mpro-inhibitor complexes.

Energy Terms	Mpro-ML188	Mpro-ECI2	Mpro-ECI3	Mpro-ECI4
Δ*E*_vdW_	−207.4 (11.1)	−210.5 (19.7)	−190.9 (12.6)	−199.6 (11.5)
Δ*E*_elec_	−33.7 (6.1)	−12.4 (3.8)	−34.0 (7.4)	−4.3 (2.4)
Δ*G*_polar_	132.3 (9.1)	104.9 (8.3)	162.2 (14.4)	99.5 (8.0)
Δ*G*_non-polar_	−20.1 (0.6)	−18.3 (0.7)	−20.3 (0.5)	−18.3 (0.59)
Δ*G*_binding_	−128.9 (11.5)	−136.3 (19.7)	−83.0 (14.1)	−122.7 (13.0)

## Data Availability

All data are contained within the article or its Appendix A as figures or tables.

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
