# Peer review of "Identification of and Mechanistic Insights into SARS-CoV-2 Main Protease Non-Covalent Inhibitors: An In-Silico Study"

_ijms, 2023, doi:10.3390/ijms24044237_

Round 1
Reviewer 1 Report
- A brief summary of the work:
- In this work, the authors used the crystal structure of Mpro of SARS-CoV-2 to design a VS protocol that was used to screen the ZINC database for potential inhibitors. Three compounds were identified and undergo a 100-ns MD simulation followed by intensive analysis to get insights about the dynamics, thermodynamics, binding free energy (BFE), and interaction energies and modes. Based on their analysis, they propose that VdW interactions is crucial for Mpro binding while H-bond interactions has to be considered carefully in regard to the desolvation energy penalty.
General concept comments:
- The work has been conducted nicely with intensive analysis of the MD simulations provided that gave useful insights for designing novel Mpro inhibitors. Figures and tables have clear descriptions in their captions. The references list includes recent work.
- Please consider the below comments:
1. State how the MD frames were selected for calculations, were the frames sampled from the entire trajectory or a certain interval? And in which frequency?
2. Indicate whether in the 2D structure or in text, the stereochemistry of the chiral centers in the zinc hits.
3. The binding modes of the zinc hits could be discussed more in terms of assigning rings/groups to each subpocket.
4. The suggestions given in the discussion could be more specific, example: clarifying in which subpocket it is recommended to increase the vdW interactions.
Specific comments:
The following typos are recommended to change:
- The title “in silico” instead of “in silicon”
- p.5 L173, L178, L181, : top-ranked pose instead of top one pose
- p22 L774: top-ranked pose instead of top one pose
- p.17 L528 This is corroborated by the overwhelming number advantage of the inter-molecular: remove advantage
Reviewer 2 Report
The paper is done according to the instructions given in the journal guidelines. Organization of paper is adequate with, material is ordered in a way that is logical, clear, and easy to follow. Author cited sources adequately and all the citations in the text are listed in the References section. Research data are presented and visualized in clear way, which all makes paper readable. Paper is well structured and well written.
Reviewer 3 Report
In the present study, the authors have performed a molecular modeling study of SARS-CoV-2 main protease to identify a noncovalent inhibitor.
1) There seems to be no novelty in this study previously several studies have been conducted where ZINC database compounds screened against the main protease e.g. Ghufran et. al (https://doi.org/ 10.3390/bioengineering10010100) screening the ZINC database compounds against main protease.
2) The authors have not provided any parameters used in the docking protocol.
3) There are several serious issues with Molecular dynamics simulations, authors have performed no control simulations without ligands, to see the effect of ligands on protein structures, and results have been claimed based on single trajectory simulations.
4) The authors should check if the systems were equilibrated correctly before production simulations, since the CA RMSD of Mpro-EC11 is very high, which indicates there is a problem with the system.
Overall, the authors should justify the novelty of this study and cite all the previous studies where the ZINC database has been screened against Mpro protein, and perform control simulations (Mpro without ligand) and each simulation should be performed in duplicate or triplicate to make any meaningful inferences from the data. Otherwise, I will not recommend this paper for publication.
Round 2
Reviewer 3 Report
The authors have provided a rebuttal, however, I am not satisfied with their reply. At least running a control simulation with just protein in water would be helpful to improve the quality of this paper. It will be beneficial to see how much ligands affect the protein which will be helpful to enhance the quality of the article. It's always good to publish work that other researchers can follow in their future studies. Incomplete work is not meaningful.
Round 3
Reviewer 3 Report
The authors have made changes suggested by me and the manuscript can be accepted in its present form.